# Counting Cohesive Subgraphs with Hereditary Properties

## ABSTRACT

The classic clique model has properties of hereditaries and cohesiveness. Here hereditaries means a subgraph of a clique is still a clique. Counting small cliques in a graph is a fundamental operation of numerous applications. However, the clique model is often too restrictive for practical use, leading to the focus on other relaxed-cliques with properties of hereditaries and cohesiveness. To address this issue, we investigate a new problem of counting general hereditary cohesive subgraphs (HCS). All subgraphs with properties of hereditaries and cohesiveness can be called a kind of HCS. To count HCS, we propose a general framework called HCSPivot, which can be applied to count all kinds of HCS. HCSPivot can count most HCS combinatorially without explicitly listing them. Two additional noteworthy features of HCSPivot are its ability to (1) simultaneously count HCS of any size and (2) simultaneously count HCS for each node or each edge. Based on our HCSPivot framework, we propose two novel algorithms with several carefully designed pruning techniques to count $s$-defective cliques and $s$-plexes, which are two specific types of HCS. We conduct extensive experiments on 8 large real-world graphs, and the results demonstrate the high efficiency and effectiveness of our solutions.

**ACM Reference Format:**

Anonymous Author(s). 2024. Counting Cohesive Subgraphs with Hereditary Properties. In *Proceedings of ACM Conference (Conference'17)*. ACM, New York, NY, USA, 15 pages. https://doi.org/10.1145/nnnnnnn.nnnnnnn

## 1 INTRODUCTION

Counting small cohesive subgraphs in a graph is a fundamental problem in graph mining. It has applications in various fields, such as community detection, network analysis, graph neural networks, and bioinformatics [7, 9, 15, 20, 40, 41, 51, 53, 59, 60, 71]. Among all subgraphs, clique is considered one of the most important due to the perfect hereditary and cohesive property. Hereditary property means that any induced subgraph of a clique is still a clique, and cohesive property means great reachability between the nodes. In the real-world, a group often has properties of cohesiveness and hereditary. For example, a group of friends should have abilities to reach each other and any subgroups are still a group of friends. Therefore, counting cliques serves as a basic operator in many applications [9, 40, 53, 59, 60, 71].

However, the constraint of clique is often too strict for real-world applications because the interaction between members of a group may not be direct, or there may be missing data in a real-world system. To overcome this limitation, many relaxed clique models

have been proposed. Notable examples include $s$-defective clique [16, 22, 72], $s$-plex [21, 23, 57, 75], $s$-clique [41], $\gamma$-quasi-clique [12, 49], $k$-core [56], $k$-truss [32, 64], and $k$-edge connected subgraph [15, 74]. In this work, we focus mainly on the relaxed clique models with both hereditary and cohesive properties, i.e. the *Hereditary Cohesive Subgraphs* (HCS). Clique, $s$-defective clique, and $s$-plex are typical types of HCS.

Specifically, $s$-defective clique is a relaxed clique and $s$-plex is a relaxed $s$-defective clique. A graph is an $s$-defective clique if there exist at most $s$ missing edges in total compared to cliques. Since the maximum count of missing edges is restricted, $s$-defective clique is typically a model of cohesive subgraph. The subgraphs of an $s$-defective clique are still $s$-defective cliques because the deletion of nodes will not lead to the increase of missing edges. Further, $s$-plex allows at most $s$ missing edges for each node, i.e., has at most $s$ non-neighbors. Similar to $s$-defective clique, $s$-plex is also hereditary because the deletion of nodes will not lead to increasing the count of non-neighbors of any node.

Instead of only counting cliques in a graph, we study a new and more general problem of counting HCS. Like clique counts, the counts of HCS have diverse applications in graph analysis. Below, we highlight two specific applications.

*Motif-based Graph Clustering.* Motif-based graph clustering has been recognized as the state-of-the-art method for detecting real-world communities in a network [9, 60]. The motif-based graph clustering method aims to minimize the so-called motif-based conductance, which is an important concept in network analysis whose definition is mainly based on the count of motifs [9, 60]. Hence, the key for motif-based graph clustering methods is to compute the count of motifs. Our experiments show that compared to the traditional clique counts [9, 60], the count of HCS, such as $s$-defective clique and $s$-plex, can improve the quality of clustering. This highlights the practical importance of the problem of counting HCS.

*Network comparison.* The graph profile [46], which is based on the count of subgraphs, shows the structural similarities or differences between graphs. This enables us to identify graph properties that are shared or distinct, contributing to comparative network analysis and classification tasks. In this work, we define a new graph profile metric based on the count of HCS. Our experiments show that such a new metric outperforms existing metrics in distinguishing different types of networks, suggesting that the counts of HCS offer great potential as a structural feature for analyzing complex networks.

**Challenges.** Although counting HCS has many practical applications, it is a challenging task. First, HCS is more complex than cliques (clique is a special kind of HCS), and the counts of HCS are often much larger than that of cliques, making HCS harder to enumerate and count. For example, on the Epinion network, the count of 8-cliques is $4.53 \times 10^8$ while the count of 1-defective clique with size 8 is $4.02 \times 10^9$ and the count of 1-plex with size 8 is $1.95 \times 10^{10}$. Second, there is no previous work for counting HCS. Although there are several algorithms for counting cliques, they are not suitable for counting other HCS like $s$-defective clique or $s$-plex. This is because

the complex structures of HCS require new search strategies, pruning techniques, and optimizations that are not present in the clique counting algorithms. As a result, new algorithms and techniques need to be developed to tackle the challenges of counting HCS.

**Contributions.** To overcome the computation challenges of counting HCS, we propose a novel pivot-based framework, called HCSPivot. HCSPivot utilizes the hereditary property to grow HCS from small to large through backtracking. HCSPivot can count most HCS combinatorially without explicitly listing them. Two useful features of HCSPivot are that (1) it can simultaneously count HCS with size in a range, and (2) it can also obtain *local count* of HCS for each node or each edge, where the local count of a node (or an edge) means the number of HCS containing that node (or edge). Based on the HCSPivot framework, we also develop two specific algorithms for counting *s*-defective cliques and *s*-plexes respectively. We design pivot-node selection strategies, $k$-core-based candidate reduction techniques, and upper-bounding pruning techniques to reduce the unpromising nodes and unnecessary search branches. The results of comprehensive experiments demonstrate the high efficiency and effectiveness of our solutions. In summary, the main contributions of this work are as follows.

*Novel* HCS *counting frameworks.* We propose a novel framework to count HCS in a graph, namely HCSPivot. HCSPivot can count HCS combinatorially. The framework is general for all HCS. HCSPivot can also locally count HCS and simultaneously count HCS with size in a given range. To our knowledge, we are the first to study the HCS counting problem and provide systematic and efficient approaches to solve this problem.

*New algorithms for s-defective clique and s-plex counting.* We propose two novel counting algorithms for the *s*-defective clique and *s*-plex counting problems based on the proposed HCSPivot framework. We develop two carefully designed pivot-node selection strategies and several non-trivial pruning techniques to boost the efficiency of these algorithms.

*Extensive experiments.* We conduct comprehensive experiments using 8 real-world large graphs. The results show that (1) both HCSList and HCSPivot are efficient when counting very small HCS. (2) HCSPivot is up to 7 orders of magnitude faster than the baseline for counting relatively large HCS. (3) HCSPivot is efficient for both locally counting in all nodes (or edges) and simultaneously counting HCS with size in a given range $[q_l, q_r]$. These results demonstrate the high efficiency of our pivot-based solutions.

For reproducibility purpose, the source code of our work is available at [4].

## 2 PRELIMINARIES

Denote by $G = (V, E)$ an undirected graph where $V$ is the set of nodes and $E \subseteq V \times V$ is the set of edges. The neighbors of each node $u \in V$ is $N(u) \triangleq \{v | (u, v) \in E\}$. The 2-hop neighbors of each node $u$ is $N_2(u) \triangleq \{w | (u, v) \in E, (w, v) \in E, (u, w) \notin E\}$. Given a graph $G(V, E)$ and a nodes set $Q \subseteq V$, define the edges induced by $Q$ as $E(Q) \triangleq \{(u, v) | (u, v) \in E, u \in Q, v \in Q\}$. We use $G(Q) = (Q, E(Q))$ to denote the subgraph induced by $Q$. If the context is clear, we replace $G(Q)$ with $Q$. For representation simplicity, we replace the size $|Q|$ with $q$. We define the total missing edges of $G(Q)$ as $\overline{m}(Q) = \binom{q}{2} - |E(Q)|$, and define the missing edges

of a specific node $u$ as $\overline{m}(u, Q) = |Q \setminus \{u\}| - |N(u) \cap Q|$. Note that $|Q \setminus \{u\}| = q - 1$ if $u \in Q$, $|Q \setminus \{u\}| = q$ otherwise.

A $k$-core of $G$ is a maximal subgraph in which every node has a degree no less than $k$ within the subgraph [56]. The core number of a node $u$ denotes the maximum $k$ such that there is a $k$-core containing $u$. The degeneracy ordering of $V$ is an ordering $\{v_1, v_2, ...\}$ such that $v_i$ has the minimum degree in the subgraph $G(\{v_i, v_{i+1}, ...\})$ [45]. A nice property of degeneracy ordering is that $\forall i, |N(v_i) \cap \{v_{i+1}, v_{i+2}, ..\}| \leq \delta$, where $\delta$ is the value of degeneracy. Note that the degeneracy value $\delta$ is equal to the maximum core number of the nodes in $G$, which is often very small in real-world networks [14, 26]. In the ordered graph, we define the (2-hop) outgoing neighbors of a node $v_i$ as $\vec{N}(v_i) = N(v_i) \cap \{v_{i+1}, v_{i+2}, ...\}$ and $\vec{N_2}(v_i) = N_2(v_i) \cap \{v_{i+1}, v_{i+2}, ...\}$.

*Definition 2.1 (Hereditary graph).* Given a graph with property $\mathcal{P}$, if all induced subgraphs also meet $\mathcal{P}$, the graph is called a hereditary graph.

Definition 2.1 gives the concept of hereditary graphs. Among the hereditary subgraphs, we are interested in *Hereditary Cohesive Subgraphs* (HCS in short), because cohesive subgraphs are often with many important practical applications in network analysis [14]. It is easy to verify that the classic clique subgraph (completed subgraph) is a kind of HCS, as clique is cohesive and any subgraph of a clique is also a clique. However, because of the strict constraint of the clique model, many relaxed clique models are often used in practical applications. In this work, we focus mainly on two widely-used relaxed clique models for cohesive subgraph, *s*-defective clique (*s*-dclique in short) [72] and *s*-plex [57], which also satisfy the hereditary property.

*Definition 2.2 (s-dclique [72]).* Given a graph $G$ and a nodes set $Q \subseteq V$, $G(Q)$ is a *s*-dclique if $\overline{m}(Q) \leq s$.

*Definition 2.3 (s-plex [57]).* Given a graph $G$ and a nodes set $Q \subseteq V$, $G(Q)$ is a *s*-plex if $\forall u \in Q, \overline{m}(u, Q) \leq s$.

It is worth mentioning that Definition 2.3 is slightly different from the traditional $k$-plex definition [57], as we exclude $u$ when defining $\overline{m}(u, Q)$. In essence, the *s*-plex is equivalent to the $(k-1)$-plex based on the traditional definition [57].

Note that clique is a special case of *s*-dclique and *s*-plex (clique is a 0-dclique and 0-plex). Both *s*-dclique and *s*-plex are not naturally cohesive as clique [57]. However, as suggested in [57], we can easily make them cohesive by restricting their diameters to no larger than 2. Fortunately, as shown in Lemma 2.4 and Lemma 2.5, very mild conditions can achieve this goal. Due to the space limits, all missing proofs are in the Appendix E.

LEMMA 2.4. *A s-dclique with size q such that $q - 2 \geq s$ has diameter at most 2.*

LEMMA 2.5 ([57]). *A s-plex with size q that $q \geq 2s + 1$ has diameter at most 2.*

For practical applications, the value of $s$ is often not very large (e.g., $s \geq 4$) [16, 23, 29, 72, 75]. When $s$ is large, the *s*-dclique and *s*-plex are likely to lose their cohesiveness. Thus, the conditions in Lemma 2.4 and Lemma 2.5 are easy to meet.

For representation simplicity, we use $(q, s)$-dclique ($(q, s)$-plex) to represent an *s*-dclique (*s*-plex) with size $q$.

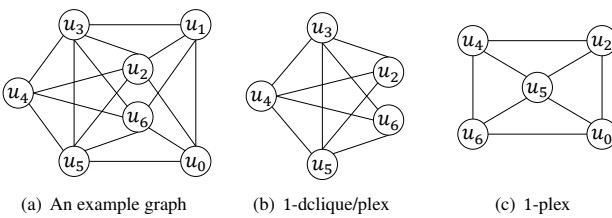

(a) An example graph     (b) 1-dclique/plex     (c) 1-plex

**Figure 1: Running example.**

---

**Algorithm 1:** The listing strategy

---

**Input:** $G = (V, E)$, two integers $q$ and $s$
**Output:** The count of hereditary cohesive subgraphs.

1   Let $V'$ be ordered by degeneracy ordering $\{v_1, v_2, ...\}$;
2   **for** $i = 1$ *to* $|V'|$ **do**
3     $C \leftarrow \vec{N}(v_i) \cup \vec{N}_2(v_i)$;
     /* Construct the candidate set     */
4     Listing($C, \{v_i\}$);
5   **Procedure** Listing($C, R$)
6     **if** $|R| = q - 1$ **then**
7       $answer \leftarrow answer + |C|$;
8       **return**;
9     Split $C$ into $C_1$ and $C_2$; /*depending on the specific problem*/
10    Listing($C_1, R$);
11    **for** $u \in C_2$ **do**
12      $C \leftarrow C \setminus \{u\}$;
13      $C' \leftarrow \{v \in C | R \cup \{u\} \cup \{v\} \, is \, an \, \text{HCS}\}$;
14      Listing($C', R \cup \{u\}$);

---

*Example 2.6.* Fig. 1(a) illustrates an example graph $G$. $G$ is a $(7, 2)$-plex and any subgraph of $G$ is a 2-plex. In Fig. 1(b), the subgraph induced by $\{u_2, u_3, u_4, u_5, u_6\}$ is both a $(5, 1)$-dclique and $(5, 1)$-plex. The subgraph induced by $\{u_0, u_2, u_4, u_5, u_6\}$ (Fig. 1(c)) is a $(5, 1)$-plex. It is easy to derive that $G$ contains 1 $(5, 1)$-dclique and 9 $(5, 1)$-plexes.

**Problem Statement.** Given a graph $G(V, E)$, parameters $q, s$ and a kind of HCS $((q, s)$-dclique or $(q, s)$-plex), our goal is to exactly count the number of HCS in $G$.

## 3   THE PIVOT-BASED SOLUTIONS

In a graph, the cliques can have complex overlaps, i.e. a sub-clique may be in many larger cliques. HCS is also similar. In real-world graphs, a small HCS can be in numerous larger HCS. For example, the sub-HCS $\{u_4, u_5, u_6\}$ is contained in both Figure 1(b) and 1(c). A natural question that arises is whether it is possible to utilize the overlaps to reduce redundant calculations. Additionally, is it possible to obtain the counts of HCS combinatorially, rather than listing each one? We propose a novel pivot-based framework that addresses these questions.

To aid understanding, we will first introduce a listing strategy for the pivot-based framework. Then, we present the pivot-based counting framework.

### 3.1   Warm up: a listing strategy

To address the potential overlap, we propose a listing strategy, which enumerates each HCS in a graph exactly once in a backtracking way.

Algorithm 1 describes the listing strategy. The algorithm first orders the graph into a degeneracy ordering (line 1), which can be computed in $O(|E|)$ time using the core decomposition algorithm [8]. In this paper, we focus mainly on the HCS with a diameter at most 2 (Lemma 2.4 and 2.5), so the initial candidate set is a set of all one-hop and two-hop outgoing neighbors (line 3). Note that a property of degeneracy ordering is that $|\vec{N}(v_i)| \leq \delta$, where $\delta$ is the value of degeneracy. Thus we can derive that $|C|$ is in the order of $O(\delta^2)$ (line 3).

Based on the degeneracy ordering, the algorithm lists each HCS on the lowest-rank node of HCS (lines 2-4), which can guarantee that each HCS is only enumerated once. Initially, for each lowest-rank node $v_i$, the Listing procedure sets $R = \{v_i\}$ and constructs the candidates set $C$ for $v_i$ (line 4). Then, for each $v_i$, the Listing procedure lists all HCS that contain $v_i$ as the lowest-rank node in the next backtracking call (lines 5-14).

The Listing procedure maintains a candidate set $C$ and a sub-HCS $R$, where each node in $C$ can be added into $R$ to form a larger sub-HCS. The Listing procedure utilizes the idea of divide-and-conquer. It divides the candidate set into two parts: one part is used directly as the candidate set for the next level of recursion, while the other part is used for listing. In Algorithm 1, the candidate set is split into two sets, $C_1$ and $C_2$ (line 5). $C_1$ is the candidate set of the next recursive call directly (line 6) and $C_2$ is to list each node (lines 7-10). For each $u$ in $C_2$, Listing enumerates all the HCS containing $u$ (lines 9-12), and removes $u$ from $C_2$ to avoid repeated enumeration (line 10). Line 11 utilizes the hereditary of HCS. When $R \cup \{u\} \cup \{v\}$ is not an HCS, $v$ should not be in the candidate set $C'$. Finally, Listing terminates when $|R| = q - 1$ because $R \cup \{u\}$ is an HCS for all $u \in C$ (lines 6-8). The following theorem shows that such a listing strategy can also correctly count all HCS.

THEOREM 3.1. *Algorithm 1 can count* HCS *correctly.*

Note that the listing strategy requires listing each HCS. The following theorem shows the worst-case time and space complexity of Algorithm 1. The time complexity of clique, $s$-dclique and $s$-plex is different (lines 9 and 13 of Algorithm 1). Let $\tau_{\text{HCS}}$ be an upper bound of that time complexity for a specific kind of HCS. The details of $\tau_{dclique}$ and $\tau_{plex}$ are in Section 4.4 and Appendix B.3 respectively.

THEOREM 3.2. *The worst-case time complexity Algorithm 1 is* $O(|V|\delta^{2(q-1)} \times \tau_{\text{HCS}})$, *and the space complexity is* $O(|V| + |E| + q\delta^2)$.

A useful feature of the listing strategy is that it classifies all sub-HCS into three categories (except the nodes contained in $R$):

  (I) the HCS only containing nodes in $C_1$;
 (II) the HCS only containing nodes in $C_2$;
(III) the HCS containing nodes both in $C_1$ and $C_2$.

With this classification, we can develop a pivot-based technique that can significantly reduce redundant calculations by computing the count of the class-III HCS combina.

### 3.2   A general pivot-based counting framework

As described previously, the HCS of class-III contains a sub-part in $C_1$. By hereditaries, this sub-part is also an HCS. The sub-HCS must

---

**Algorithm 2:** Pivot-based counting framework (HCSPivot)

---

1 **Procedure** Listing($C, R, D$)
2    **if** $|R| = q - 1$ **then**
3       $answer \leftarrow answer + |D| + |C|$;
4       **return**;
5    **if** $|C| = 0$ **then**
6       $answer \leftarrow answer + \binom{|D|}{q-|R|}$;
7       **return**;
8    Split $C$ into $C_1$ and $C_2$; /*depending on the specific problem*/
9    **if** *there exists a pivot node $u_p$ in $C_2$ (Definition 3.3)* **then**
10       $C_2 \leftarrow C_2 \setminus \{u_p\}$;
11       Listing($C_1, R, D \cup \{u_p\}$);
12    **else**
13       Listing($C_1, R, D$);
14    **for** $u \in C_2$ **do**
15       $C \leftarrow C \setminus \{u\}$;
16       $C' \leftarrow \{v \in C | R \cup \{u\} \cup \{v\} \text{ is an HCS}\}$;
17       Listing($C', R \cup \{u\}, D$);

---

be in class-I because it only has nodes in $C_1$. Note that the recursive call in line 6 of Algorithm 1 has already searched all class-I HCS, thus it is no need to repeatedly list such sub-HCS when counting the class-III HCS. Inspired by this, we propose a novel pivoting technique.

*Definition 3.3 (pivot node).* Let $\mathbb{H}$ be the set of HCS in $C_1$ that $\forall H \in \mathbb{H}, R \cup H$ *is an* HCS. A node $u_p \in C_2$ can serve as a pivot node if $\forall H \in \mathbb{H}, R \cup H \cup \{u_p\}$ is an HCS.

From $\mathbb{H}$, we can obtain the set of HCS of class-III $\{H \cup \{u_p\} | H \in \mathbb{H}\}$ without actually listing them. It is straightforward to deduce that the number of HCS of size $q + 1$ in $\{H \cup \{u_p\} | H \in \mathbb{H}\}$ is equivalent to the number of HCS of size $q$ in $\mathbb{H}$.

Based on this idea, we propose a new counting framework, which is detailed in Algorithm 2. In addition to the candidate set $C$ and the sub-HCS $R$, the new Listing procedure in Algorithm 2 maintains a node set $D$. The set $D$ contains all the pivot nodes selected so far. Initially, $D = \emptyset$ and the candidate set $C$ is divided into two sets $C_1$ and $C_2$ (line 8). If there exists a node $u_p$ that can serve as a pivot node, it is removed from $C_2$ and added to $D$ (lines 9-11). If there is no pivot node, the process proceeds in the same manner as Algorithm 1 (line 13). Once the size of $R$ reaches $q - 1$, each node in $D$ and $C$ can be added to $R$ to generate a new HCS (lines 2-3). When $C$ is empty, every $q - |R|$ nodes in $D$ combined with $R$ form a distinct HCS (lines 5-6).

By replacing the Listing procedure in Algorithm 1 with the Listing procedure outlined in Algorithm 2, we obtain a new general framework, called HCSPivot. Below, we analyze the correctness of HCSPivot.

**Correctness of the HCSPivot framework.** The backtracking process of Algorithm 2 can be represented as a recursion tree, where the root node has $D = \emptyset$ and the leaves have either $|R| = q - 1$ or $|C| = 0$. Each HCS lies on exactly one path of the recursion tree. Let us label the recursive calls of Listing in line 11, line 13, and line 17 as $L_1$, $L_2$, and $L_3$, respectively. Each node in the recursion tree either has a child node from $L_1$ or $L_2$ and multiple child nodes

from $L_3$. If an HCS only consists of nodes in $C_1$, it will be in the path down to either $L_1$ or $L_2$ according to whether the pivot node exists. If an HCS contains a pivot node $u_p$ and all other nodes are in $C_1$, it will be in the path down to $L_1$. Finally, if an HCS contains nodes in $C_2$ (excluding $u_p$ if it exists, in line 10), it will be in the path down to $L_3$. These include all cases. Each HCS can occur in only one of them. Thus, we can claim that HCSPivot is capable of accurately computing the count of HCS. Theorem 3.4 formally proves that HCSPivot is correct.

THEOREM 3.4. HCSPivot *correctly counts the* HCS.

Note that if no pivot node is selected in each recursion of Algorithm 2, Algorithm 2 degenerates to Algorithm 1. Thus, the time complexity of HCSPivot is the same as that of in Theorem 3.2.

However, for the two specific HCS considered in this paper, i.e., *s*-dclique and *s*-plex, we can select valid pivot nodes in most recursions (for *s*-dclique counting, we can also guarantee that the pivot node always exists in each recursion), which can boost the performance of HCSPivot. Indeed, as shown in our experiments, HCSPivot can be up to 7 orders of magnitude faster than Algorithm 1 based on such a powerful pivoting technique.

## 3.3 Discussions

**Counting HCS with size in a range $[q_l, q_r]$ simultaneously.** A striking feature of Algorithm 2 is that it is highly adaptable and capable of simultaneously computing the counts of HCS of different sizes in a range of $[q_l, q_r]$. This is achievable because the recursion tree of Algorithm 2 is only slightly impacted by the parameter $q$. By making a few modifications to the algorithm, we can count HCS with sizes in the range $[q_l, q_r]$ simultaneously. These modifications include: (1) modifying $|R| = q - 1$ in line 2 to $|R| = q_r - 1$, and (2) modifying $answer \leftarrow answer + \binom{|D|}{q-|R|}$ in line 6 to $answer_q \leftarrow answer_q + \binom{|D|}{q-|R|}$ for every $q \in [q_l, q_r]$. In the experiments, we will use the counts of HCS with various sizes to build a *graph profile* [30, 68] for a given network. The experimental results show the ability of the proposed *graph profile* to characterize different types of networks.

**Local counting.** Another important feature of Algorithm 2 is that it can obtain the local count of HCS for all nodes or edges. Here local count refers to the number of HCS that contain a specific node or edge. For a given node $u$ or edge $(u, v)$, we define its local count as $c_u$ or $c_{(u,v)}$, respectively. To implement local counting using Algorithm 2, we only need to add a counter for each node (or each edge) in line 6. More specifically, to compute the local counts for nodes, we partition the nodes into two categories: $u \in R$ and $u \in D$. If $u \in R$, then $c_u$ is updated as $c_u \leftarrow c_u + \binom{|D|}{q-|R|}$. On the other hand, if $u \in D$, then $c_u$ is updated as $c_u \leftarrow c_u + \binom{|D|-1}{q-|R|-1}$. To calculate the local counts for edges, we divide the edges into three categories: (1) $(u \in R, v \in R)$, (2) $(u \in R, v \in D)$, and (3) $(u \in D, v \in D)$. For category (1), $c_{(u,v)}$ is updated as $c_{(u,v)} \leftarrow c_{(u,v)} + \binom{|D|}{q-|R|}$. For category (2), $c_{(u,v)}$ is updated as $c_{(u,v)} \leftarrow c_{(u,v)} + \binom{|D|-1}{q-|R|-1}$. Finally, for category (3), $c_{(u,v)}$ is updated as $c_{(u,v)} \leftarrow c_{(u,v)} + \binom{|D|-2}{q-|R|-2}$. In the experiments, we will apply the local counts to construct a matrix $M$, where $M_{ij}$ is the count of HCS containing the edge $(i, j)$. Such

a matrix $M$ is then used for graph clustering applications which achieves much better performance compared to the state-of-the-art graph clustering methods.

**Relation to the pivot-based $q$-clique counting algorithm.** PIVOTER is the state-of-the-art pivot-based $q$-clique counting algorithm [34]. Actually, our HCSPivot is exactly the PIVOTER algorithm for counting $q$-clique. Specifically, select the maximum-degree node in $C$ as the pivot node $v_p$, set $C_1 = N(v_p) \cap C$ in Algorithm 2, and then we get the PIVOTER algorithm exactly. However, our HCSPivot is not only a naive extension of PIVOTER. Compared to PIVOTER, which can only be applied to count $q$-cliques, Algorithm 2 is a general framework which is capable of processing all HCS counting problems. Moreover, the idea of splitting the candidate set into $C_1$ and $C_2$ in Algorithm 2 is not used in PIVOTER [34], which is critical to develop a general approach to all HCS counting problems. Based on Algorithm 2, we need to design different candidate set partition and pivot node selection strategies when handling various HCS counting problems.

**Extension to larger diameter.** To apply HCSPivot to count hereditary subgraphs with diameter larger than 2, we can simply modify line 3 of Algorithm 1 to include candidates in longer hops.

## 4 PIVOT-BASED $S$-DCLIQUE COUNTING

For applying HCSPivot to count $s$-dcliques, we need to solve : (1) how to split $C$ into $C_1$ and $C_2$, and (2) how to choose a pivot node.

Lemma 4.1, derived from Definition 3.3, provides the criteria for a node $u$ to be a pivot node for $s$-dclique counting.

LEMMA 4.1. *Let $\mathbb{H}$ be the set of $s$-dcliques in $C_1$ such that $\overline{m}(R \cup H) \leq s$ for each $H \in \mathbb{H}$, and $u$ be a node in $C_2$. If $\overline{m}(R \cup H \cup \{u\}) \leq s$ for each $H \subseteq \mathbb{H}$, then $u$ is a pivot node.*

### 4.1 A basic pivoting technique

By Lemma 4.1, it is easy to see that if $\overline{m}(u, R \cup H) = 0$ for all $H \in \mathbb{H}$, the node $u$ must be a pivot node because $\overline{m}(R \cup H \cup \{u\}) = \overline{m}(R \cup H) + \overline{m}(u, R \cup H) \leq s$. Based on this observation, a basic pivoting method is to select a node $u$ that $\overline{m}(u, R) = 0$ and $\overline{m}(u, H) = 0$. We can select such pivot node $u$ from the common neighbors of $R$ and then set $C_1$ as $N(u) \cap C$. Here we select the pivot node from common neighbors of $R$ to make $C_1$ as large as possible. This is because a large $C_1$ will result in a small $C_2$ which can reduce the number of recursive branches in the recursion tree of Algorithm 2, thus improving the efficiency of the algorithm. However, such a straightforward method has two limitations. First, it requires the pivot must be a common neighbor of $R$ which is very restrictive and such a pivot may not exist (the common neighbor of $R$ may be empty). Second, even though the pivot node has the maximum degree among the common neighbors of $R$, $|C_2|$ may still be large, meaning that there will be many recursive branches generated by the nodes in $C_2$, which may also slow down the algorithm.

### 4.2 An improved pivoting technique.

To overcome the limitations of the basic pivoting method, we propose an improved pivoting technique that can ensure that the pivot node always exists in each recursion. Specifically, we relax the restriction $\forall H \subseteq \mathbb{H}, \overline{m}(R \cup H \cup \{u\}) \leq s$ in Lemma 4.1 to $\forall H \subseteq \mathbb{H}, \overline{m}(H \cup \{u\}) \leq s$. This change means that we do not consider the non-neighbors

of $u$ in $R$. Instead of choosing the pivot node from the common neighbors of $R$, we directly select the pivot node from $C$ to let the size of $C_1 = N(u) \cap C$ be the largest. This may result in the problem of $\exists u \in D, \overline{m}(u, R) > 0$ at the leaf of the recursion tree (lines 5-7 of Algorithm 2). Recall that for the basic pivoting technique, it has $\forall u \in D, \overline{m}(u, R) = 0$ since $u$ is chosen from the common neighbors of $R$. Thus, we can choose arbitrary $q - |R|$ nodes from $D$. However, now it may occurs the case that $\overline{m}(R \cup H) > s$ where $|H| = q - |R|, H \subseteq D$. We need to compute how many subsets with size $q - |R|$ in $D$ are valid answers. Fortunately, this is not a complicated issue, since $G(D)$ is always a clique as described in the following Theorem 4.2.

THEOREM 4.2. *If the node $u$ is selected as the pivot node and let $C_1 = N(u) \cap C$, $G(D)$ must be a clique for the parameter $D$ in each recursion node of Algorithm 2.*

With our improved pivoting technique, the task of choosing arbitrary $q - |R|$ nodes from $D$ when $|C| = \emptyset$ (as stated in line 6 of Algorithm2) transforms into computing the number of subsets $H$ with $H \subseteq D$ and $|H| = q - |R|$ such that $H \cup R$ is a valid $(q, s)$-dclique.

THEOREM 4.3. *If $\sum_{u \in H} \overline{m}(u, R) \leq s - \overline{m}(R)$ holds, $H \cup R$ is a valid $(q, s)$-dclique for $H \subseteq D, |H| = q - |R|$.*

As a consequence, the problem of counting all valid $(q, s)$-dcliques is equivalent to a classic variant of *0-1 Knapsack Problem*, where the knapsack size is $s - \overline{m}(R)$ and the item weight is $\overline{m}(u, R)$ for each $u \in D$. This can be solved in $O(s|D|^2)$ time and $O(s|D|)$ space using *Dynamic Programming* [42].

The correctness of the pivot-based $s$-dclique counting algorithm can be guaranteed by Theorem 3.4 and Theorem 4.2.

Due to the space limits, we provide an example of HCSPivot for $s$-dclique counting in Appendix A.

### 4.3 Pruning techniques for $s$-dclique counting

Our first pruning technique is designed to reduce the candidate set $C$ before enumeration. Specifically, we aims to reduce the size of $\vec{N}(v_i) \cup \vec{N}_2(v_i)$ (line 3 of Algorithm 1). Below, we present two useful lemmas that will be used to reduce $\vec{N}(v_i)$ and $\vec{N}_2(v_i)$ respectively.

LEMMA 4.4. *For each node $u \in \vec{N}(v_i)$, if $u$ and $v_i$ are contained in a $(q, s)$-dclique, we have $|\vec{N}(u) \cap \vec{N}(v_i)| \geq q - s - 2$.*

Lemma 4.4 shows that for an $(q, s)$-dclique that contains $v_i$, the subgraph induced by all the nodes in $\vec{N}(v_i)$ must be a $(q-s-2)$-core. As a result, for each $v_i$ in line 3 of Algorithm 1, we can reduce $\vec{N}(v_i)$ by computing the $(q - s - 2)$-core on the subgraph induced by $\vec{N}(v_i)$.

LEMMA 4.5. *For each node $u \in \vec{N}_2(v_i)$, if $u$ and $v_i$ are contained in a $(q, s)$-dclique, $u$ has at least $q - s - 1$ neighbors in $\vec{N}(v_i)$.*

By Lemma 4.5, we can further prune the candidate set $C$ by removing the nodes that have less than $q - s - 1$ neighbors in the $(q - s - 2)$-core of $\vec{N}(v_i)$ from $\vec{N}_2(v_i)$.

Our second pruning technique is to eliminate unnecessary branches of the procedure Listing in advance. However, how do we determine if a branch of Listing is unnecessary without actually accessing it? Below, we propose an upper-bounding technique to solve this issue.

Assuming the procedure Listing is in the state between line 15 and line 16 of Algorithm 2, where $u$ is already removed from $C$. We need to determine whether the following branch (line 16) can

be pruned based on the $u$, $R$, and $C$. Let the nodes in $N(u) \cap C$ be ordered according to the count of non-neighbors in $R$, from smallest to largest, as $\{v_1, v_2, ...\}$. It follows that $\overline{m}(v_i, R) \leq \overline{m}(v_{i+1}, R)$. We can define $\omega(u, R, C) = \max_i\{\sum_{j \leq i} \overline{m}(v_j, R) \leq s - \overline{m}(R \cup \{u\})\}$. Since $\{v_1, v_2, ...\}$ is ordered, $\omega(u, R, C)$ is an upper bound on the number of nodes in $N(u) \cap C$ that can be added into $R \cup \{u\}$. With this information, we can calculate the total upper bound $\gamma_d(u, R, C)$ which can be used to prune the branch in advance if $\gamma_d(u, R, C) < q$. Lemma 4.6 provides further details on this process.

LEMMA 4.6. *Let* $\gamma_d(u, R, C) = |R \cup \{u\}| + \min\{s - \overline{m}(R \cup \{u\}), \overline{m}(u, C)\} + \omega(u, R, C)$. *Then,* $\gamma_d(u, R, C)$ *is an upper bound on the size of the s-dclique that the enumeration branch of Algorithm 2 can potentially reach.*

It is worth remarking that the work that focused on the maximum $s$-dclique problem [29] also defines an upper bound. However, their upper bound uses $|N(u) \cap C|$ directly instead of our $\omega(u, R, C)$. Thus, our upper bound can be deemed as a tighter bound which may also be more effective than the one defined in [29] for solving the maximum $s$-dclique problem.

**Implementations.** In the implementation of the $s$-dclique counting algorithm, we maintain an array $A$ of size $|V|$ that $A[v]$ stores the value of $\overline{m}(v, R)$. When $u$ is added into $R$, for each $v \in C' \setminus N(u)$, $A[v]$ should increase 1. Correspondingly, $A[v]$ should decrease 1 after the removal of $u$ from $R$ (after line 17 of Algorithm 2). The maintenance of $A$ costs linear time. With the help of the array $A$, we can get the count of non-neighbors of each node in constant time.

The computation of the upper bound $\omega(u, R, C)$ does not sort the set $N(u) \cap C$. Instead, we construct a bucket array $B$ where $B[i]$ is the count of nodes in $N(u) \cap C$ that have $i$ non-neighbors in $R$. Based on $B$, the time complexity of the computation of $\omega(u, R, C)$ is only $O(s)$.

## 4.4 Complexity analysis

According to Theorem 3.2, the time complexity of counting dclique depends on the cost of each recursion tree node, i.e. $\tau_{dclique}$. Theorem 4.7 presents a bound of $\tau_{dclique}$.

THEOREM 4.7. $O(\tau_{dclique}) = O(\delta^4)$.

The practical performance of our pivot-based algorithm is extremely faster than the worst-case bound thanks to the benefits of the new pivot node selection strategy: (1) by selecting the node with the maximum degree in $C$ as the pivot node, the size of $C_1 = N(u_p) \cap C$ is maximized and the number of nodes to list is minimized; (2) pivot-nodes always exist at each node of the full recursion tree, and thus more $s$-dcliques are counted in a combinational way.

Example A.1 in Appendix A illustrates the running process of our pivot-based $s$-dclique counting algorithm.

**Pivot based $s$-plex counting.** Due to space limits, the details of the application of our HCSPivot framework for $s$-plex counting are in Appendix B. Similarly, we also develop pivot node choosing strategies, candidate set size reduction techniques, and upper-bound-based prune techniques to boost efficiency.

## 5 EXPERIMENTS

In this section, we conduct extensive experiments to evaluate the performance of the proposed solutions. In addition, we also evaluate

**Table 1: Datasets**

| Networks | $|V|$ | $|E|$ | $\delta$ | Type |
|---|---|---|---|---|
| WikiV | 7115 | 201524 | 53 | Social network |
| Caida | 26475 | 106762 | 22 | Autonomous system network |
| Epinion | 75879 | 811480 | 67 | Social network |
| EmailEu | 265009 | 728960 | 37 | Communication network |
| Amazon | 403394 | 4886816 | 10 | Communication network |
| DBLP | 425957 | 2099732 | 113 | Co-authorship network |
| Pokec | 1632803 | 44603928 | 47 | Social network |
| Skitter | 1696415 | 22190596 | 111 | Autonomous system network |

the effectiveness of our algorithms by presenting case studies and demonstrating their applications in Section 5.3 and Appendix C.1.

## 5.1 Experimental setup

**Algorithms.** We evaluate 4 algorithms, namely Dlist, Plist, Dpivot, and Ppivot. Dlist and Plist are listing-based algorithms (Algorithm 1) to count $s$-dclique and $s$-plex respectively. Dpivot and Ppivot are pivot-based algorithms (Algorithm 2) to count $s$-dclique and $s$-plex respectively. All algorithms are implemented in C++ and all of them are integrated with the pruning techniques developed in Section 4.2 and Appendix B.2 as their default setting. Since this is the first work to study the problem of HCS counting, we use the list-based algorithms Dlist and Plist as the baselines.

**Datasets.** We selected 8 real-world networks across different domains to evaluate the performance of different algorithms. The details of the datasets can be found in Table 1. All the datasets used in our experiments are obtained from the SNAP project [37].

All the experiments are conducted on a server with an AMD3990X CPU, 256GB memory, and Linux CentOS 7 operating system.

**The choice of $s$.** Similar to previous studies on enumerating maximal $s$-dcliques [22] and $s$-plexes [23, 66, 75], we mainly evaluate different HCS counting algorithms with $1 \leq s \leq 3$. The reasons are as follows. First, real-world applications often require that the subgraph pattern is cohesive and also very similar to clique [23, 66, 75]. When $s > 3$, the HCS might be sparse, thus it is more reasonable to consider a small $s$ as done in previous work [23, 66, 75]. Second, for a large $s$, the problem becomes more challenging due to the exponential growth in the search space and the count as $s$ increases. In fact, even for maximal $s$-plex enumeration problem (often much easier than $s$-plex counting), existing solutions are confined to instances with $s \leq 3$ [23, 66, 75]. We also do not consider $s = 0$ ($q$-clique counting), because our HCSPivot degrades to the state-of-the-art $q$-clique counting algorithm PIVOTER, as discussed in Section 3.3.

## 5.2 Performance studies

**Comparing the listing and pivot-based algorithms.** Comparing the performance of Dlist and Dpivot in Table 2, we observe that they exhibit similar performance when $q$ is small. For example, on WikiV, Dlist takes 10.2 seconds to count $(5, 1)$-dclique, while Dpivot needs 9.62 seconds. However, as $q$ becomes larger, Dpivot can be several orders of magnitude faster than Dlist on many networks. For example, on DBLP, when $s = 1$ and $q = 9$, Dlist consumes 162301.4 seconds, while Dpivot takes only 0.1 seconds. We also find that on some networks, such as Caida and Amazon, both Dlist and Dpivot perform similarly even for a large $q$. This is because the count of HCS with size $q$ in each network differs. If the count is large, Dlist is often much slower than Dpivot, while if the count is not very large, Dlist performs comparably with Dpivot. For instance, Epinion

**Table 2: Running time (sec) of different algorithms with various $q$ and $s$ ( '-' means the algorithm can not terminate in a day )**

| Networks | Running time (sec) | | | | | | | | | | | | | | | | | |
|---|---|---|---|---|---|---|---|---|---|---|---|---|---|---|---|---|---|---|
| | s = 1 | | | | | | s = 2 | | | | | | s = 3 | | | | | |
| | s-dclique | | | s-plex | | | s-dclique | | | s-plex | | | s-dclique | | | s-plex | | |
| | $q$ | Dlist | Dpivot | $q$ | Plist | Ppivot | $q$ | Dlist | Dpivot | $q$ | Plist | Ppivot | $q$ | Dlist | Dpivot | $q$ | Plist | Ppivot |
| WikiV | 5 | 10.2 | **9.6** | 5 | 18.3 | **16.7** | 7 | **126.4** | 134.9 | 7 | 4315.5 | **3139.3** | 10 | 3083.8 | **720.3** | 15 | - | **70049.5** |
| | 10 | 13.1 | **5.7** | 10 | 80.9 | **14.6** | 12 | 60.1 | **43.5** | 12 | 5059.8 | **1618.3** | 15 | 95.3 | 115.4 | 20 | 3612.8 | **1548.9** |
| | 15 | **1.4** | 1.8 | 15 | 8.1 | **2.4** | 17 | **3.7** | 7.8 | 17 | - | **88.9** | 20 | **3.9** | 9.4 | 25 | **1.9** | 2.1 |
| Caida | 6 | 0.2 | **0.1** | 6 | 0.3 | **0.2** | 7 | 0.9 | **0.6** | 7 | 19.3 | **9.8** | 7 | 22.9 | **6.6** | 7 | - | **14131.6** |
| | 9 | 0.1 | **0.0** | 9 | 0.3 | **0.1** | 9 | 0.6 | **0.3** | 9 | 12.9 | **5.4** | 9 | 3.6 | **2.0** | 9 | - | **375.8** |
| | 12 | **0.0** | 0.0 | 12 | 0.1 | **0.0** | 12 | **0.1** | 0.1 | 12 | 4.3 | **1.1** | 12 | 0.7 | **0.4** | 12 | 193.0 | **90.7** |
| Epinion | 5 | 48.2 | **36.0** | 5 | **62.6** | 66.3 | 7 | 1450.8 | **588.9** | 7 | - | **14507.9** | 10 | - | **3747.7** | 25 | - | **93791.8** |
| | 10 | 320.7 | **25.8** | 10 | 1813.7 | **109.6** | 12 | - | **267.9** | 12 | - | **26011.0** | 15 | - | **1137.0** | 30 | - | **733.7** |
| | 15 | 462.2 | **12.9** | 15 | 7951.5 | **56.9** | 17 | 6013.3 | **92.6** | 17 | - | **12645.6** | 20 | 1046.6 | **240.7** | 35 | 1.0 | **0.9** |
| EmailEu | 5 | 3.4 | **1.9** | 5 | 4.0 | **3.7** | 7 | 16.8 | **14.8** | 7 | 386.5 | **281.4** | 10 | 225.7 | **48.1** | 10 | - | **14966.8** |
| | 10 | 1.9 | **0.7** | 10 | 11.4 | **1.7** | 9 | 36.7 | **4.4** | 12 | 508.0 | **113.6** | 15 | 244.7 | **6.6** | 15 | - | **3577.9** |
| | 15 | **0.2** | 0.2 | 15 | 0.8 | **0.3** | 12 | 6.2 | **3.5** | 12 | 27.1 | **7.7** | 20 | 118.8 | **0.4** | 20 | - | **77.5** |
| Amazon | 6 | 2.8 | **1.0** | 6 | 4.2 | **2.7** | 7 | 6.0 | **1.7** | 7 | 39.5 | **18.3** | 7 | 53.1 | **10.5** | 7 | - | **23008.3** |
| | 9 | 0.7 | **0.3** | 9 | 0.9 | **0.5** | 9 | 1.5 | **0.5** | 9 | 7.1 | **2.5** | 9 | 5.3 | **1.1** | 9 | 254.7 | **106.7** |
| | 12 | 0.3 | **0.1** | 12 | **0.1** | 0.1 | 12 | 0.3 | **0.1** | 12 | **0.2** | 0.2 | 12 | 0.4 | **0.2** | 12 | 2.6 | **1.0** |
| DBLP | 5 | 4.9 | **0.5** | 5 | 6.1 | **0.9** | 7 | 1100.2 | **0.7** | 7 | 1518.9 | **11.8** | 7 | 1439.1 | **5.0** | 10 | - | **930.6** |
| | 7 | 1005.5 | **0.2** | 7 | 1192.8 | **0.3** | 12 | - | **0.2** | 12 | - | **2.8** | 9 | 182028.8 | **1.8** | 15 | - | **550.6** |
| | 9 | 162301.4 | **0.1** | 9 | 193757.0 | **0.2** | 17 | - | **0.2** | 17 | - | **2.3** | 12 | - | **1.3** | 20 | - | **459.3** |
| Pokec | 5 | 279.6 | **139.7** | 5 | **493.8** | 538.6 | 7 | 797.2 | **493.1** | 7 | 66895.4 | **8683.0** | 10 | - | **936.9** | 15 | - | **54975.3** |
| | 10 | 125.3 | **22.4** | 10 | 468.2 | **57.1** | 12 | 1209.2 | **67.6** | 12 | 16931.4 | **1584.5** | 20 | 2603.2 | **14.5** | 20 | - | **4817.3** |
| | 15 | 163.8 | **10.0** | 15 | 673.6 | **13.7** | 17 | 1075.8 | **14.8** | 17 | 14930.8 | **238.3** | 30 | 30.5 | **4.8** | 25 | - | **119.8** |
| Skitter | 10 | - | **4209.7** | 30 | - | **54752.0** | 30 | - | **25884.6** | 55 | - | **49106.6** | 40 | - | **67878.3** | 60 | - | **-** |
| | 30 | - | **1849.2** | 40 | - | **13224.9** | 40 | - | **7784.1** | 60 | - | **1069.3** | 50 | - | **7355.9** | 65 | - | **2177.8** |
| | 50 | - | **906.7** | 50 | - | **728.8** | 50 | - | **1201.2** | 65 | - | **96.3** | 60 | - | **668.9** | 70 | - | **136.2** |

has $1.7 \times 10^9$ $(15, 1)$-dcliques, whereas Caida only has $3.2 \times 10^4$ $(12, 1)$-dcliques. Similarly, Ppivot is more efficient than Plist on complex networks when $q$ is large. These results demonstrate the high efficiency of the proposed pivot-based counting algorithms.

**Difference on counting $s$-dclique and $s$-plex.** From Table 2, we observe that Plist is slower than Dlist, and Ppivot is slower than Dpivot for a given $s$ and $q$. For instance, on Skitter when $s = 1$ and $q = 30$, Dpivot is an order of magnitude faster than Ppivot. This result suggests that $s$-plex is a more complex structure than the $s$-dclique, which is often more difficult to count.

Dpivot **and** Ppivot **with various $q$ and $s$.** In Table 2, we observe that the running time of Dpivot and Ppivot decreases as $q$ increases. However, the counter-intuitive thing is that the count of HCS may not decrease as $q$ becomes larger. For instance, Skitter has $9.3 \times 10^{13}$ $(10, 1)$-dcliques and $1.1 \times 10^{21}$ $(30, 1)$-dcliques, but Dpivot runs faster on counting $(30, 1)$-dcliques than $(10, 1)$-dcliques. This phenomenon occurs because larger values of $q$ can prune more candidate sets, thanks to the core-based and upper-bound based pruning techniques developed in Section 4.2 (and Appendix B.2). Note that the size of the search tree in Algorithm 2 is almost not affected by the parameter $q$. Thus, no matter what the value of $q$ is, Algorithm 2 will enumerate all the large HCSs. Therefore, the larger the value of $q$, the smaller the size of the candidate set, and the faster Dpivot and Ppivot are.

In Table 2, we can also observe that the running time of Dpivot and Ppivot increases as the value of $s$ increases. For instance, on the WikiV network, Dpivot takes 9.62 seconds for $(5, 1)$-dclique and 134.9 seconds for $(7, 2)$-dclique, which is an increase of 14×. On the other hand, Ppivot takes 16.7 seconds for $(5, 1)$-plex and 3139.3 seconds for $(7, 2)$-plex, which is an increase of 188×. This is because the number of HCSs in a network increases significantly when the parameter $s$ increases.

**Table 3: The reduction rate of candidate pruning technique.**

| $q$ | Networks | The reduced ratio of the candidate set | | | | | |
|---|---|---|---|---|---|---|---|
| | | s = 1 | | s = 2 | | s = 3 | |
| | | s-dclique | s-plex | s-dclique | s-plex | s-dclique | s-plex |
| 10 | WikiV | 95.33% | 95.39% | 94.72% | 94.02% | 93.95% | 91.52% |
| | Epinion | 95.81% | 95.96% | 95.30% | 94.92% | 94.64% | 92.81% |
| | Amazon | 94.83% | 93.94% | 94.07% | 93.41% | 93.28% | 92.06% |
| | Pokec | 97.66% | 97.72% | 97.36% | 97.28% | 97.01% | 95.89% |
| 20 | WikiV | 97.59% | 97.67% | 97.52% | 97.44% | 97.39% | 97.19% |
| | Epinion | 97.62% | 97.66% | 97.55% | 97.51% | 97.45% | 97.28% |
| | Amazon | 100% | 100% | 100% | 100% | 100% | 100% |
| | Pokec | 99.80% | 99.72% | 99.71% | 99.47% | 99.60% | 99.06% |

**Evaluating the pruning technique: candidate reduction.** Table 3 reports the reduction rate of the candidate pruning technique. $q$ is fixed to 10 and 20. The reduction rate is computed as $\frac{c_{pre} - c_{now}}{c_{pre}}$, where $c_{pre}$ is the total size of all the candidate set before pruning, i.e. $\sum_{v_i} |\vec{N}(v_i) \cup \vec{N}_2(v_i)|$ in line 3 of Algorithm 1 and $c_{now}$ is the total size after pruning. The table shows that the candidate reduction technique can remove over 90% of unnecessary vertices, which is a significant speed-up. When $q$ goes larger, more vertices can be removed from the candidate set. For example, when $s = 1$ and $q = 10$, the total candidate set size of WikiV after pruning on $s$-dclique counting is $3.5 \times 10^5$. When $s = 1$ and $q = 20$, the size becomes $2.0 \times 10^4$. Therefore, the candidate reduction based pruning technique is effective.

**Evaluating the prune technique: upper bound.** In Fig. 2, we compare Ppivot against Ppivot-nup, and compare Dpivot against Dpivot-nup on WikiV, where Ppivot-nup and Dpivot-nup are Algorithm 2 without the upper-bound based prune techniques. Similar results can also be observed on the other datasets. As shown in Fig. 2, the effectiveness of the upper bounds increases when $q$ becomes larger. The effect of $s$ on the effectiveness of the upper bounds also increases with increasing $s$. For instance, by fixing $q = 20$, Ppivot-nup is 4× slower than Ppivot when $s = 1$ while 47× when $s = 2$.

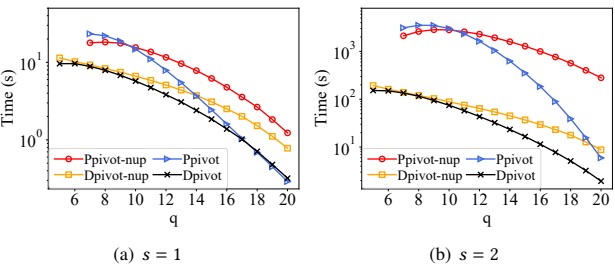

Figure 2: Effectiveness of the upper bounds (WikiV).

**Table 4: The percentage of HCSs counted in combination**

| q | Networks | s-dclique | | s-plex | |
|---|---|---|---|---|---|
| | | $s = 1$ | $s = 2$ | $s = 1$ | $s = 2$ |
| 7 | WikiV | 95.7% | 91.8% | 87.6% | 57.4% |
| | Epinion | 92.8% | 87.3% | 84.3% | 51.8% |
| | Amazon | 100.0% | 100.0% | 100.0% | 98.4% |
| | Pokec | 98.5% | 96.5% | 94.8% | 72.1% |
| 14 | WikiV | 100.0% | 100.0% | 100.0% | 99.8% |
| | Epinion | 100.0% | 100.0% | 100.0% | 99.6% |
| | Amazon | 100.0% | 100.0% | 100.0% | 100.0% |
| | Pokec | 100.0% | 100.0% | 100.0% | 100.0% |

**The effectiveness of the pivoting technique.** In Algorithm 2, the HCSs are counted by listing (line 3) or counted in a combinatoric manner (line 6). Clearly, the more HCSs are counted in a combinatoric manner, the better the acceleration of the pivot-vertex technique. Table 4 shows the percentage of HCSs counted in a combinatoric manner. In Table 4, Dpivot has larger percentages than Ppivot. Moreover, Ppivot is more sensitive to the parameter $s$. This is because Dpivot always has a pivot vertex at each node of the recursion tree, while Ppivot depends on $R$, $C$ and $s$ (as stated by Corollary B.3). Table 4 also shows that the HCSs with larger size are more easy to count in a combinatoric way. With larger value of $q$, the condition $|R| = q - 1$ (line 2 in Algorithm 2) is harder to meet. In general, our pivot-based algorithms enable a large number of HCSs to be counted in a combinatoric manner. These results further confirm the high efficiency of the proposed pivot-based solutions.

**Additional experiments.** Due to the space limits, more experiments on memory cost, counting HCS with size in $[q_l, q_r]$ simultaneously, and local counting are presented in Appendix C.

## 5.3 Application 1 : HCS-based graph clustering

In this section, we show the application of the HCS counts for motif-based graph clustering. One of the most popular metrics in community detection is *conductance* [9, 28, 38, 54, 60, 69]. The conductance is the ratio of the count of edges leaving the community and the count of edges in the community. Similarly, the motif-based conductance [9, 60] is the ratio of the count of motifs leaving the community and the count of motifs in the community, which is formulated as $\Phi(S) = \frac{C_M(S:\bar{S})}{min(C_M(S), C_M(\bar{S}))}$, where $\bar{S}$ is the remainder vertices of $S$, $C_M(S)$ is the count of the motif $M$ in $G(S)$ and $C_M(S : \bar{S})$ is the count of the motif $M$ that contains both vertices in $S$ and $\bar{S}$. $\Phi(S)$ measures how well a community $S$ preserves the occurrences of $M$ compared to its complement $\bar{S}$. A community with low motif-based conductance means that it preserves the occurrences of the motif $M$ well within the community.

With our HCSPivot, we can get the weight matrix $W_M \in N^{|V| \times |V|}$ where $(W_M)_{ij}$ is the count of motifs containing edge $(i, j)$. By the

**Table 5: The results of HCS-based graph clustering**

| Methods | Metrics | | | |
|---|---|---|---|---|
| | ARI | Purity | NMI | $F_1$ |
| $(3, 1)$-dclique/plex | **0.47** | **0.67** | **0.68** | **0.49** |
| $(4,1)$-dclique | 0.34 | 0.62 | 0.62 | 0.38 |
| $(4,1)$-plex | 0.33 | 0.63 | 0.63 | 0.36 |
| 3-clique | 0.30 | 0.64 | 0.64 | 0.33 |
| 4-clique | 0.23 | 0.59 | 0.59 | 0.27 |
| Louvain | 0.33 | 0.45 | 0.58 | 0.38 |
| pSCAN | 0.02 | 0.28 | 0.28 | 0.10 |
| LPA | 0.10 | 0.52 | 0.49 | 0.13 |
| Infomap | 0.28 | 0.52 | 0.62 | 0.33 |

local counting property of HCSPivot as discussed in Section 3.3, the computation of the weight matrix is very efficient. Using this weight matrix, we can follow the same steps as the classic spectral clustering algorithms [9, 40, 60] to obtain clusters that minimize the motif-based conductance. These classic spectral clustering algorithms use the clique count, while we use HCS to devise an HCS-based spectral clustering algorithm.

Table 5 presents a comparison of the clustering performance of our proposed HCS-based spectral clustering algorithm against several state-of-the-art graph clustering algorithms, including clique-based (3-clique and 4-clique) spectral clustering [9, 60], Louvain [6], pSCAN [13], label propagation algorithm (LPA) [61], and Infomap [25, 52]. We implement our spectral clustering algorithms using the commonly-used scikit-learn Python package [48]. For other popular algorithms, we use their open-source implementations that have been thoroughly tested [6, 13, 25, 48, 52]. We evaluate the algorithms on the email-Eu-core network downloaded from the SNAP project [37], which contains 42 ground truth communities. Based on the ground-truth communities, we can apply 4 widely-used metrics, including ARI [62], Purity [55], NMI [67], and $F_1$ score, to evaluate the clustering performance of different algorithms. As shown in Table 5, our $(3, 1)$-dclique or $(3, 1)$-plex model (when $q = 3$ and $s = 1$, $(3, 1)$-dclique and $(3, 1)$-plex are identical) achieves the best clustering performance with all 4 metrics. In general, both HCS and clique-based spectral clustering algorithms (the first 5 rows) significantly outperform the other baselines to identify real-world communities, and our HCS-based solutions can further outperform clique-based spectral clustering algorithms [9, 60]. These results demonstrate the high effectiveness of the proposed solutions in community detection applications.

Due to space limits, the Application 2 of utilizing the count of HCS for network characterization is in Appendix C.1, and the related works is put in Appendix D.

## 6 CONCLUSION

In this work, we address a new problem of counting hereditary cohesive subgraphs (HCS) in a graph. To tackle this problem, we develop a novel pivot-based framework which counts most HCS combinatorially without the need for exhaustive listing. We devise specific algorithms with several carefully designed pruning techniques to count the $s$-defective cliques and $s$-plexes in a graph. We conduct comprehensive experiments to evaluate the efficiency of our algorithms. The results reveal that the pivot-based solutions exhibit significantly higher performance, being up to 7 orders of magnitude faster than the listing-based solution. In addition, we also evaluate the performance of our methods in graph clustering and network characterization applications, and the results demonstrate the high effectiveness of the proposed solutions.

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

# A  EXAMPLE OF $s$-DCLIQUE COUNTING

*Example A.1.* Fig. 3(a) depicts a sub-tree of the total recursion tree generated by Algorithm 2 on counting the $(4, 1)$-dcliques that include node $u_0$. The calls to Listing in line 11, line 13, and line 17 of Algorithm 2 are labeled as $L_1$, $L_2$, and $L_3$, respectively. The root node has a candidate set of $C = \{u_1, u_2, \ldots, u_6\}$, a sub-dclique of $R = \{u_0\}$, and an empty set $D = \{\}$. The node $u_3$ is selected as the pivot node because it has the maximum degree. The pivot node is then added to the set $D$ and the candidate set $C$ becomes $\{u_1, u_2, u_4, u_5, u_6\}$. The node $u_2$ has the maximum degree in the updated candidate set, so it is processed through $L_1$ and the node $u_6$ is processed through $L_3$. The last step is to solve a *0-1 Knapsack Problem* at the leaf node. Let us consider the example of the leaf node $R = \{u_0\}$, $D = \{u_2, u_3, u_4, u_5\}$. The problem is to choose $q - |R| = 3$ nodes from $D$. Since $G(D)$ must be a clique, we need to ensure that the three chosen nodes have at most $s - \overline{m}(R) = 1$ non-neighbors in $R$. Clearly, the two correct answers are $\{u_2, u_3, u_5\}$ and $\{u_2, u_4, u_5\}$.

# B  PIVOT-BASED $S$-PLEX COUNTING

Compared to counting $s$-dcliques, counting $s$-plexes is a more complicated task as each node is allowed to miss at most $s$ edges.

## B.1  The pivoting technique

Lemma B.1, which derives from Definition 3.3, provides the criteria for a node $u$ to be considered as a pivot node.

LEMMA B.1. *Let $\mathbb{H}$ be the set of s-plex in $C_1$ that $\forall H \in \mathbb{H}, \forall v \in R \cup H, \overline{m}(v, R \cup H) \leq s$. Let $u$ be a node that $u \in C_2$ and $H'$ represents $R \cup H \cup \{u\}$ in short. If $\forall H \subseteq \mathbb{H}, \forall v \in H', \overline{m}(v, H') \leq s$, $u$ is a pivot node.*

The condition $\forall H \subseteq \mathbb{H}, \forall v \in H', \overline{m}(v, H') \leq s$ means that for any $H \in \mathbb{H}$, adding the node $u$ into the set $R \cup H$ always results in an $s$-plex. It encompasses both (1) the requirement that $\overline{m}(u, H') \leq s$, meaning that $u$ itself has at most $s$ non-neighbors, and (2) the requirement that $\forall v \in (R \cup H) \setminus N(u), \overline{m}(v, H') \leq s$, meaning that the non-neighbors of $u$ still have at most $s$ non-neighbors after the addition of $u$. Based on these observations, we propose a pivot-selection technique for counting $s$-plexes based on Theorem B.2.

THEOREM B.2. *Let $u \in C$, and let $C_1 = N(u) \cap C$. The node $u$ can serve as a pivot node if $\forall v \in R \setminus N(u), \overline{m}(v, R \cup C_1) \leq s - 1$.*

PROOF. Note that $H' = R \cup H \cup \{u\}$ can be considered as $R \cup H$ with the addition of $\{u\}$. For every $v \in R \cap N(u)$, it still has a maximum of $s$ missing edges after $u$ is added to $R \cup H$. For $v \in R \setminus N(u)$, $\overline{m}(v, H') \leq s$ is ensured because $\overline{m}(v, R \cup C_1) \leq s - 1$. And for $v \in H$, since $u$ and $v$ are neighbors, $\overline{m}(v, R \cup H) = \overline{m}(v, H')$. Putting all it together, all nodes in $H'$ have at most $s$ missing edges, and thus $u$ is a valid pivot node.  □

According to Theorem B.2, the steps of choosing the pivot node is as follows. First, for each $u \in C$ and $C_1 = N(u) \cap C$, we evaluate whether all nodes in $R \setminus N(u)$ have at most $s - 1$ non-neighbors in $R \cup C_1$. Then, we obtain a set of nodes that can serve as the pivot node. Among them, we choose the one that has maximum degree in $C$ as the pivot node. If no node in $C$ can serve as the pivot node, we set $C_1 = C \cap N(v)$ that $v$ has maximum degree in $C$.

However, for each $u \in C$, $C_1 = N(u) \cap C$ and for each $w \in R \setminus N(v)$, computing the count of non-neighbors of $w$ in $R \cup C_1$ is quite a heavy work. To enhance the efficiency, we improve the

strategy of choosing the pivot node according to Corollary B.3, which comes from Theorem B.2.

COROLLARY B.3. *Let $u$ be a node in $C$ and let $C_1 = N(u) \cap C$. $u$ can serve as a pivot node if $\forall v \in R \setminus N(u), \overline{m}(v, R \cup C) \leq s - 1$.*

The correctness of Corollary B.3 can be ensured by Theorem B.2 because $C_1$ is a subset of $C$. According to Corollary B.3, we just need to compute the count of non-neighbors in $R \cup C$ for each node in $R$, which can be done in linear time. Similarly, among the set of nodes that can serve as the pivot node, we choose the one that has maximum degree in $C$ as the pivot node, and let $C_1$ be the set of neighbors of the pivot node. It is important to note that if the condition "$\forall v \in R \setminus N(u), \overline{m}(v, R \cup C) \leq s - 1$" does not hold, there is no pivot node can be selected. In this case, we set $C_1 = N(u) \cap C$ where $u$ is the maximum-degree node in $C$ (line 13 of Algorithm 2).

## B.2  Pruning techniques for $s$-plex counting

We use $k$-core based pruning technique to reduce the candidate size for $s$-plex counting. Specifically, we can first compute a $(q - s - 2)$-core on the subgraph induced by $\vec{N}(v_i)$ without missing any $s$-plex, based on the results established in [21].

LEMMA B.4 ([21]). *Assume that $u$ and $v$ are two nodes in a $(q, s)$-plex, then $u$ and $v$ have at least $q - 2s - 2$ common neighbors if $(u, v)$ is an edge in the graph, and have at least $q - 2s$ common neighbors otherwise.*

With Lemma B.4, we can remove unpromising nodes from the candidate set $C$ in line 3 of Algorithm 1. Specifically, we first reduce $\vec{N}(v_i)$ to a $(q - s - 2)$-core by removing the nodes $\{u | u \in \vec{N}(v_i), |\vec{N}(u) \cap \vec{N}(v_i)| < q - s - 2\}$. Then, we delete the nodes in $\vec{N}_2(v_i)$ that have degree less than $q - 2s$ in the $(q - s - 2)$-core. Finally, we combine the remaining nodes to get the updated candidate set without losing any $(q, s)$-plex.

To reduce the enumeration branches, we devise an upper bound on the size of the $s$-plex that the current branch can access. If this upper bound is less than $q$, we can prune the current branch as it will not produce a valid $(q, s)$-plex. Our upper bound is inspired by [23]. Specifically, when the Listing procedure is in the state between line 15 and line 16 of Algorithm 2, where the algorithm tries to enumerate the plexes containing a node $u$, we calculate an upper bound which is defined as $\gamma_p(u, R, C) = |R \cup \{u\}| + \max_i\{\sum_{j \leq i} \overline{m}(v_j, R) \leq \sum_{v \in R}(s - \overline{m}(v, R))\} + \min(s - \overline{m}(u, R), \overline{m}(u, C))$, where $\{v_1, v_2, \ldots\}$ is the set $N(u) \cap C$ ordered according to $\overline{m}(v_i, R)$. This upper bound contains three parts: (1) $|R \cup \{u\}|$, the size of the listed sub-plex, (2) $\max_i\{\ldots\}$, the upper bound of the count of neighbors of $u$ that can be added into the listed sub-plex, and (3) $\min(s - \overline{m}(u, R), \overline{m}(u, C))$, the upper bound of the count of non-neighbors of $u$ that can be added into the listed sub-plex. It is easy to derive that $\gamma_p(u, R, C)$ is valid upper bound.

**Implementations.** In the implementation of the listing-based $s$-plex counting algorithm, we maintain an array $As$ with size $|V|$. For $v \in C$, $As[v]$ is exactly the set of nodes $R \setminus N(v)$. Since the count of the non-neighbors of $v$ is at most $s$, $As$ takes at most $O(s|V|)$ space. Similar to $s$-dclique, the maintenance of $As$ takes $O(|C|)$ time for each $u \in C$. Specifically, $u$ should be inserted into $As[v]$ for $v \in C \setminus N(u)$ if $u$ is added into $R$ and removed otherwise. With the array $As$, we can get the non-neighbors of each node efficiently.

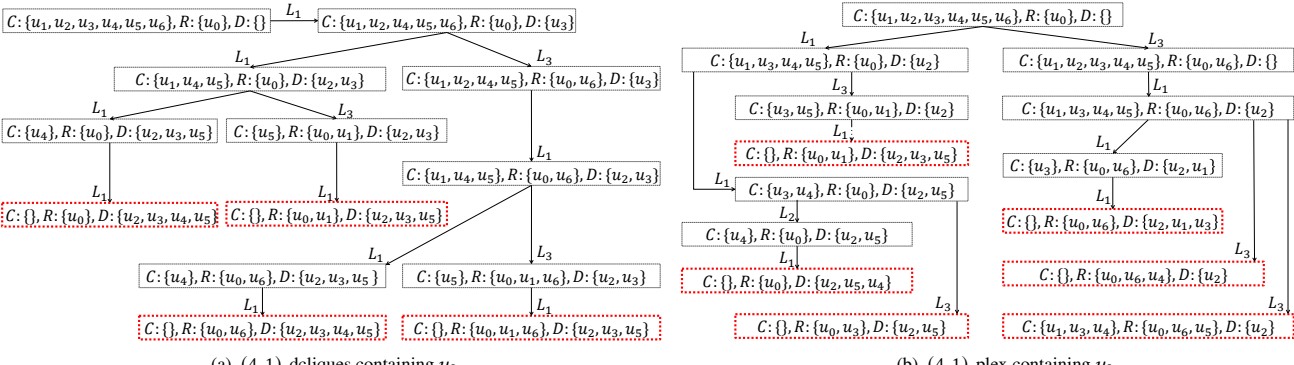

(a) $(4,1)$-dcliques containing $u_0$

(b) $(4,1)$-plex containing $u_0$

**Figure 3: Illustration of the recursion tree of Algorithm 2 on the graph in Fig. 1(a).**

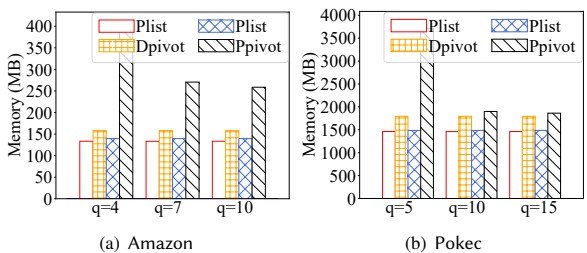

(a) Amazon

(b) Pokec

**Figure 4: Memory overheads of different algorithms**

Note that an efficient implementation of checking whether $R \cup \{u\} \cup \{v\}$ is a $s$-plex (line 16 of Algorithm 2) is nontrivial. A straightforward implementation is that for each $v \in C$ and for each $w \in R \cup \{u\} \setminus N(v)$, we verify whether $\overline{m}(w, R \cup \{u\} \cup \{v\}) \le s$. The drawback of this implementation is that it needs to check the non-neighbors of each node in $C$. To improve this, we observe that for each $w \in R \cup \{u\} \setminus N(v)$, only the nodes that $\overline{m}(w, R \cup \{u\}) = s$ do not meet the condition because $w$ and $v$ are non-adjacent. We can claim that $w$ is also a non-neighbor of $u$. To see this, we assume to the contrary that $w \in N(u)$, then we have $\overline{m}(w, R) = s$. This is impossible because $w$ is the non-neighbor of $v \in C$ and $\overline{m}(w, R \cup \{v\}) = s + 1$ which is contradictory to the fact that $R \cup \{v\}$ is a $s$-plex. Therefore, our improved implementation is that for each $w \in R \setminus N(u)$, if $\overline{m}(w, R \cup \{u\}) = s$, we remove the non-neighbors of $w$ in $C$. Note that in this improved implementation, it is no need to check the non-neighbors of the nodes in $C$.

## B.3 Complexity Analysis

For the $s$-plex counting algorithm, the number of the recursion-tree nodes is also consistent with Theorem 3.2. Theorem B.5 shows the time complexity taken in each tree node for $s$-plex counting.

THEOREM B.5. $O(\tau_{plex}) = O(s\delta^4)$.

PROOF. According to the analysis of the implementation above, checking whether $R \cup \{u\} \cup \{v\}$ is a $s$-plex takes at most $O(s|C|^2)$ time. The maintenance of $As$ consumes linear time for each node $u \in C$. Thus, the total cost of maintaining $As$ is $O(|C|^2)$. Putting it all together, the total time complexity in each recursion-tree node of the listing-based $s$-plex counting algorithm is $O(s|C|^2) = O(s\delta^4)$. □

By Theorem 4.7 and B.5, $s$-plex counting algorithm is more sensitive to the parameter $s$ than $s$-dclique counting. As confirmed

in our experiments (see Table 2), with the increase of $s$, the running time of the $s$-plex counting algorithm increases more significantly compared to the $s$-dclique counting algorithm.

*Example B.6.* Fig. 3(b) is the recursion tree of Algorithm 2 on counting $(4,1)$-plex. We label the calls of Listing in line 11, line 13, line 17 of Algorithm 2 as $L_1, L_2, L_3$, respectively. The root node has $C = \{u_1, u_2, ..., u_6\}$ and $R = \{u_0\}$. Among the root node, $u_2, u_5$ and $u_6$ can serve as the pivot node and they all have 4 neighbors in $C$. The nodes $u_1, u_3$ and $u_4$ cannot serve as the pivot node. This is because they have non-neighbors in $C \cup R$ and the count of the non-neighbor in $C \cup R$ is at most $s - 1 = 0$ according to Corollary B.3. Suppose that choose $u_2$ as the pivot node, and then $C_1 = N(u_2) = \{u_1, u_3, u_4, u_5\}, C_2 = \{u_6\}$. $C_1$ is the candidate set of the child node through $L_1$. The node $u_6$ is inserted into $R$ in the child node through $L_3$. Note that on the recursion node with $C = \{u_3, u_4\}$, $R = \{u_0\}$, and $D = \{u_2, u_5\}$, no node can serve as the pivot node. In this case, we set $C_1 = C \cap N(u_3) = \{u_4\}$ and $C_2 = \{u_3\}$. Here a child node through $L_2$ with candidate set $C_1$ occurs, and $u_3$ is inserted into $R$ in the child node through $L_3$.

In Fig. 3(b), the red dotted leaves are the answers. The recursion node with $C = \{u_1, u_3, u_4\}$, $R = \{u_0, u_6, u_5\}$, and $D = \{u_2\}$ is the leaf node because $|R| = q - 1$ (line 2 of Algorithm 2). In the red dotted leaves, the combination of any $q - |R|$ nodes in $D$ with $R$ forms a $(4,1)$-plex. For instance, in the leaf node with $R = \{u_0, u_6\}$ and $D = \{u_2, u_1, u_3\}$, every two nodes of $D$ combined with $R$ will result in a $(4,1)$-plex.

## C ADDITIONAL EXPERIMENTS

**Memory overheads.** Fig. 4 compares the memory costs of the proposed algorithms on Amazon and Pokec given that $s = 1$. For other parameter settings and other datasets, the results are consistent. The pivot-based approaches require more memory compared to the listing-based solutions. This is because the recursive depth of the pivot-based algorithms is often deeper than the listing-based solutions, thus resulting in more space usages. However, we can also note that the memory overheads of the pivot-based solutions are still within the same order of magnitude as listing-based algorithms, due to the fact that both the listing-based and pivot-based methods are depth-first search algorithms which are typically space-efficient. For example, when $q = 4$, Plist consumes 140 MB on Amazon and Ppivot uses 394 MB on the same datasets.

**Table 6: The running time of counting** HCS **with size in** $[q_l, q_r]$ **simultaneously** ($q_l = 5, q_r = 20$)**.**

| Networks | Running time (sec) | | | |
|---|---|---|---|---|
| | Dpivot | | Ppivot | |
| | $q_l$ | $[q_l, q_r]$ | $q_l$ | $[q_l, q_r]$ |
| WikiV | 9.62 | 11.11 | 16.68 | 32.27 |
| Epinion | 35.98 | 47.595 | 66.31 | 184.17 |
| Amazon | 1.02 | 1.86 | 4.72 | 4.74 |
| Pokec | 139.73 | 146.88 | 538.55 | 570.89 |

**Table 7: The running time of locally counting**

| q | Networks | Running time (sec) | | | | | |
|---|---|---|---|---|---|---|---|
| | | Dpivot | | | Ppivot | | |
| | | global | vertex | edge | global | vertex | edge |
| 7 | WikiV | 11.3 | 11.9 | 21.7 | 29.6 | 30.1 | 58.4 |
| | Epinion | 43.3 | 45.5 | 94.5 | 119.6 | 121.8 | 288.9 |
| | Amazon | 0.8 | 0.8 | 1.2 | 1.8 | 1.8 | 2.2 |
| | Pokec | 60.6 | 65.2 | 109.1 | 188.5 | 190.3 | 268.4 |
| 14 | WikiV | 3.0 | 4.7 | 7.0 | 4.7 | 4.6 | 6.8 |
| | Epinion | 19.2 | 28.1 | 72.5 | 86.4 | 86.5 | 194.5 |
| | Amazon | 0.1 | 0.1 | 0.1 | 0.1 | 0.1 | 0.1 |
| | Pokec | 11.6 | 14.1 | 22.5 | 18.6 | 18.6 | 27.2 |

**Counting** HCS **with size in** $[q_l, q_r]$ **simultaneously.** As described in Section 3.3, Dpivot and Ppivot can count HCS with size in a range simultaneously. Table 6 shows the running time, where the column $q_l$ means the running time of counting only HCS with size $q_l$. The parameters are set as $s = 1$, $q_l = 5$ and $q_r = 20$. As shown in Table 6, The running time of counting $[q_l, q_r]$ and counting only $q_l$ are within the same order of magnitude. The results with other parameters are similar. This is because counting $[q_l, q_r]$ and counting only $q_l$ has the same size of the search tree and only has difference in the leaves nodes (as described in Algorithm 2 and Section 3.3). These results further demonstrate the advantages of the pivot-based algorithms, compared to the list-based algorithms.

**Local counting.** We compared the running time of global and local counting in Table 7, with parameters $s = 1$. Local counting includes counting HCSs in each vertex and edge. In Table 7, we find that the local-vertex counting is slightly slower than the globally counting. This is because the computation of local-vertex counting only requires scanning the vertices, as described in Section 3.3. However, the local-edge counting needs to scan the edges, and since the number of edges is larger than the number of vertices, the running time of local-edge counting is slower but still within the same order of magnitude. For example, in Table 7 with $q = 7$, local-edge counting is at most 2.42× slower than global counting. These results indicate that Dpivot and Ppivot are very efficient in locally counting HCSs for each vertex or edge.

### C.1 Application 2 : HCS-based network analysis

The graph profile (GP) is a kind of characteristic vector of given networks, which has a lot of applications in network analysis [3, 30, 43, 44, 68]. A nice property of GP is that the networks in the same domain often have similar GPs [46, 71].

The subgraph ratio profile (SRP) is a well-known and widely used GP [46]. The SRP is a vector that quantifies the relative significance of a sequence of subgraphs, denoted by $\{M_1, M_2, ...\}$, where the $i_{th}$ entry in the vector corresponds to the normalized count of a specific subgraph $M_i$. This normalization involves subtracting the count of $M_i$ in a randomly generated null graph from that of the original graph, followed by a scaling [46]. In our study, we use a null graph model based on the algorithm proposed in [36], which preserves

the core number of the original graph. The sequence of subgraphs $\{M_1, M_2, ...\}$ is composed of six types of connected subgraphs with a size of four, following the methodology of [46].

**HCS-based graph profile.** We introduce a novel graph profile, termed the HCS-based graph profile (HGP), which is constructed based on the count of HCS. The HGP is a vector where the $i_{th}$ entry represents the ratio between the count of cliques and HCSs with size $q_i$. Since cliques are a special type of HCS, this ratio can be interpreted as the probability of an HCS being a clique, which characterizes the convergence behavior of the graphs. Since HCSPivot can count HCSs with size in a range $[q_l, q_r]$ simultaneously (Section 3.3), we can compute HGP efficiently.

In Fig. 5, we compare the SRPs and HGPs on the networks in three domains [37]. We set the sequence of size of HGP as $\{4, 5, ..., 20\}$.

Fig. 5(a) plots the SRPs and HGPs on 3 Amazon networks. Ama0312, Ama0505 and Ama0601 have exactly the same shapes of HGPs and SRPs. HGPs of the networks in the same domain have the same change tendency. The same performance of SRP and HGP implies that the proposed HGP can characterize the network properties in the same domain as SRP.

Fig. 5(b) plots the SRPs. Fig. 5(c) plots dclique-based HGPs. Fig. 5(d) plots plex-base HGPs. There are 6 collaboration networks and 3 social networks. When the networks are in different domains, we find that HGP performs better than SRP. In Fig. 5(b), SRP cannot separate the collaboration and social networks. However, in Fig. 5(c) and Fig. 5(d), HGP separates the two kinds of networks clearly. As a result, HGP is a better GP than SRP when there exist networks in multiple domains. These results demonstrate the high effectiveness of the proposed HGP to characterize network properties.

## D RELATED WORK

**Subgraph Counting.** Our work is related to the subgraph counting problem which includes general subgraph counting and specific subgraph counting [51]. The recent exact algorithms for general subgraphs counting [2, 31, 47, 50] can compute the count efficiently, but they always have a small size constraint. To overcome the hardness, sampling-based approximate algorithms [1, 10, 35, 65] for general subgraph counting are well studied. Recently, machine learning techniques are also applied for subgraph counting [17, 63, 73]. However, these method are often not very accurate and also cannot provide a theoretical guarantee of the estimated counts.

Despite the general algorithms, there are a lot of algorithms designed for counting specific important subgraphs, where the most representative one is $k$-clique. The first algorithm for $k$-clique counting was introduced by [18], which is a listing-based solution. To improve the efficiency, some ordering based algorithms [24, 39] are designed. Instead of listing, Jain and Seshadhri [34] propose an elegant algorithm PIVOTER, which can count all $k$-cliques without listing them, based on the classic maximal clique enumeration algorithm [11]. There also exist several approximate clique counting algorithms [33, 70], which are often more efficient but cannot obtain the exact counts. All these algorithms are tailed to the problem of $k$-clique counting and cannot be applied to the general HCS counting problem. In addition, we also note that the complexity of the hereditary subgraph counting problem was investigated in [27]. However, no specific counting algorithm was proposed in [27].

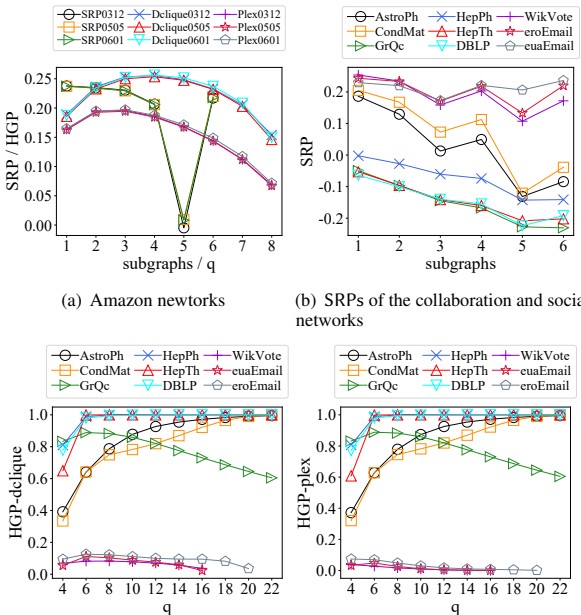

(a) Amazon newtorks

(b) SRPs of the collaboration and social networks

(c) dclique-based HGPs of collaboration and social networks

(d) plex-based HGPs of collaboration and social networks

**Figure 5: (1) Fig. 5(a) are** SRPs **and** HGPs **on the Amazon networks. The Amazon0312, Amazon0505 and Amazon0601 have the same** SRPs **and** HGPs**. (2) Fig. 5(b), Fig. 5(c) and 5(d) are** SRPs **and** HGPs **on 6 collaboration (from AstroPh, CondMat to DBLP) and 3 social (from WikVote to eroEmail) networks. In Fig. 5(b), the networks in the two domains have similar** SRPs**. In Fig. 5(c) and Fig. 5(d), the networks in the same domain still have similar** HGPs**, but the** HGPs **of the networks in the two domains are different.**

**Maximal Hereditary Subgraph Enumeration.** Our work is also related to the problem of maximal hereditary subgraph enumeration, the goal of which is to enumerate all maximal hereditary subgraphs. The widely studied maximal hereditary subgraph model is the maximal clique. The most popular algorithm for maximal clique enumeration is the classic pivot-based Bron-Kerbosch algorithm [11, 58]. Since clique model is often too restrictive, several relaxed clique models include $s$-defective clique and $s$-plex are proposed [16, 22, 23, 29, 57, 72, 75], which also satisfy the hereditary property. Recently, several pivot-based solutions for enumerating maximal $s$-defective cliques and maximal $s$-plex were proposed [22, 23, 66], which generalize the classic pivoting technique proposed for maximal clique enumeration [58]. We also note that [19] developed a general algorithm to enumerate all maximal hereditary subgraph based on the reverse search framework [5]. However, this algorithm is often much slower than the pivot-based algorithms when processing real-world graphs. All the above mentioned algorithms are tailored to maximal hereditary subgraph enumeration, and they cannot be used for HCS counting. This is because an HCS can be located in many maximal hereditary subgraphs; and it is very difficult to reduce the repeated counts when using the maximal hereditary subgraphs to count HCS.

## E  MISSING PROOFS

The proof of Lemma 2.4.

PROOF. Let $G'$ be a $s$-dclique with size $q$ such that $q - 2 \geq s$. Assume to the contrary that the length of the shortest path between $u$ and $v$ in $G'$ is 3. Clearly, it has $N(u) \cap N(v) = \emptyset$, i.e. there are no edge between $N(u)$ and $v$, and between $N(v)$ and $u$. Suppose that $u$ and $v$ have $t$ common non-neighbors in $G'$ in total. Then, the number of missing edges in $G'$ is at least $|N(u)| + |N(v)| + 1 + 2t$, i.e., $s \geq |N(u)| + |N(v)| + 1 + 2t$. Note that the total number of nodes in $G'$ is $q = |N(u)| + |N(v)| + 2 + t$. Thus, we have $s \geq q - 1$, which is a contradiction. □

The proof of Theorem 3.1.

PROOF. It is easy to check that if an HCS is in $R \cup C_1$, it will be accessed in the recursive call in line 6. If an HCS is in $R \cup C_2$ or $R \cup C_2 \cup C_1$, it will be accessed in the recursive call in line 10. As a result, no HCS will be missed by Algorithm 1. □

The proof of Theorem 3.2.

PROOF. Since the graph is ordered by degeneracy ordering, the size of a candidate set $C$ in line 3 is $O(\delta^2)$. The depth of the recursion tree of the Listing procedure is exactly $q - 1$ (line 6). Therefore, the total size of a recursion tree is $O(\delta^{2(q-1)})$. There are $|V|$ such recursion trees and the time cost of each tree node is $\tau_{HCS}$. At last, the time complexity of Algorithm 1 is $O(|V|\delta^{2(q-1)} \times \tau_{HCS})$.

For the space complexity, the main cost is the $O(\delta^2)$ space for storing the candidate set $C$ at each recursion tree node in the Listing procedure. Clearly, the total space needed for this is $O(q\delta^2)$, and the space required for storing the input graph is $O(|V| + |E|)$. As a result, the overall space complexity of the algorithm is $O(|V|+|E|+q\delta^2)$. □

The proof of Theorem 3.4.

PROOF. Without loss of generality that we randomly choose an HCS $H$ with size of $q$. We prove that $H$ is counted exactly once by HCSPivot. We set the root node of the recursion tree of Algorithm 2 as layer 0, and set the recursion tree node as layer $i$ if there is an $i$-length path from the root. In the recursion tree node, we label the recursive calls of Listing in line 11, line 13, line 17 of Algorithm 2 as $L_1, L_2, L3$, respectively. For a node of layer $i$ with parameters $C_i, R_i, D_i$, we say that the node holds the state $\mathcal{S}_i$ if $R_i \subseteq H$ and $H \subseteq R_i \cup C_i \cup D_i$. Clearly, the root node holds $\mathcal{S}_0$. Then we need to prove the transitivity that only one child node of layer $i + 1$ holds $\mathcal{S}_{i+1}$ if the current node of layer $i$ holds $\mathcal{S}_i$.

If there exist a pivot node $u_p$ and $H$ only contains nodes in $C_1 \cup \{u_p\}$, the parameters of layer $i + 1$ through $L_1$ that $C_{i+1} = C_1, R_{i+1} = R_i$ and $D_{i+1} = D_i \cup \{u_p\}$ clearly meet that $H \subseteq R_{i+1}, H \subseteq R_{i+1} \cup C_{i+1} \cup D_{i+1}$, i.e., layer $i + 1$ holds $\mathcal{S}_{i+1}$. For the other child nodes through $L_3$, the parameter $R_{i+1} = R_i \cup \{u\}$ (line 17 of Algorithm 2). Since $u \in C_2$ and $H$ only contains nodes in $C_1 \cup \{u_p\}$, it has $R_{i+1} \not\subseteq H$. Thus, only one child node holds $\mathcal{S}_{i+1}$.

If there is no pivot node $u_p$ and $H$ only contains nodes in $C_1$, the parameters of layer $i + 1$ through $L_2$ that $C_{i+1} = C_1, R_{i+1} = R_i$ and $D_{i+1} = D_i$ clearly meet that $H \subseteq R_{i+1}, H \subseteq R_{i+1} \cup C_{i+1} \cup D_{i+1}$, i.e., layer $i + 1$ holds $\mathcal{S}_{i+1}$. For the other child nodes through $L_3$, the parameter $R_{i+1} = R_i \cup \{u\}$ (line 17 of Algorithm 2). Since $u \in C_2$ and $H$ only contains nodes in $C_1$, it has $R_{i+1} \not\subseteq H$. Therefore, only one child node holds $\mathcal{S}_{i+1}$.

At last, we analyze the case that $H$ contains nodes in $C_2$ when the pivot node $u_p$ does not exist and the case that $H$ contains nodes in $C_2 \setminus \{u_p\}$ when the pivot node $u_p$ exists. Let us consider the first node $u'$ that $u' \in H$ in line 14. The child nodes through $L_3$ before $u'$ must have $R_{i+1} \nsubseteq H$ because $u'$ is the first node that $u' \in H$. The child nodes through $L_3$ after $u'$ must have $H \nsubseteq R_{i+1} \cup C_{i+1} \cup D_{i+1}$ because $u'$ is already removed from $C$ in line 15. The child node through $L_1$ or $L_2$ also has $H \nsubseteq R_{i+1} \cup C_{i+1} \cup D_{i+1}$ because $u'$ is in $C_2$. Thus, only the child node that has $R_{i+1} = R_i \cup \{u'\}$ holds the state $\mathcal{S}_{i+1}$.

With layer 0 holding $\mathcal{S}_0$ and the transitivity, the proof of the theorem is completed by induction. More specifically, when $|C| = 0$, it has $R \subseteq H$ and $H \subseteq R \cup D$. Thus $H$ contains $q - |R|$ nodes in $D$ (line 6 of Algorithm2). □

The proof of Theorem 4.2.

PROOF. Since $C_1 = N(u) \cap C$, the set of candidates $C$ is made up of the common neighbors of $D$ at all recursion nodes. Therefore, $G(D)$ must always be a clique. □

The proof of Theorem 4.3.

PROOF. Since $D$ is a clique by Theorem 4.2, i.e. $\overline{m}(H) = 0$ and $R$ has already missed $\overline{m}(R)$ edges, the missing edges in $R \cup H$ is $\overline{m}(R \cup H) = \overline{m}(R) + \overline{m}(H) + \sum_{u \in H} \overline{m}(u, R) \leq s$. □

The proof of Lemma 4.4.

PROOF. Assume $G(Q)$ is a $(q, s)$-dclique that contains $u$ and $v_i$. Let $n_{2hop} = |Q \cap \vec{N}_2(v_i)|$, and we have $|Q \cap \vec{N}(v_i)| = q - n_{2hop} - 1$. Since the maximum number of missing edges is $s$, and $v_i$ already has $n_{2hop}$ missing edges, the maximum number of missing edges of

$u$ in $\vec{N}(v_i)$ is $s - n_{2hop}$, i.e., $\overline{m}(u, Q \cap \vec{N}(v_i)) \leq s - n_{2hop}$. Therefore, $|\vec{N}(u) \cap \vec{N}(v_i)| \geq |\vec{N}(u) \cap (Q \cap \vec{N}(v_i))| = |Q \cap \vec{N}(v_i)| - 1 - \overline{m}(u, Q \cap \vec{N}(v_i)) \geq q - s - 2$. □

The proof of Lemma 4.5.

PROOF. Let $Q$ be a $(q, s)$-dclique that contains $u$ and $v_i$. Since $u \in \vec{N}_2(v_i)$, $u$ and $v_i$ is disconnected in $Q$ by definition. Suppose to the contrary that $u$ and $v_i$ have $q - s - 2$ common neighbors. Then, there are $s$ non-common neighbors in total for $u$ and $v_i$, which means that $s$ edges will be missed for $u$ and $v_i$. Since $u$ and $v_i$ is disconnected, $Q$ misses at least $s + 1$ edges, which is a contradiction. □

The proof of Lemma 4.6.

PROOF. As previously explained, $\omega(u, R, C)$ is the upper bound of the number of nodes in $N(u) \cap C$ that can be added into $R \cup \{u\}$. Similarly, it is easy to see that $\min\{s - \overline{m}(R \cup \{u\}), \overline{m}(u, C)\}$ is the upper bound of the number of nodes in $C \setminus N(u)$ that can be added into $R \cup \{u\}$. By adding these upper bounds together, we get the total upper bound represented by $\gamma_d(u, R, C)$. □

The proof of Theorem 4.7.

PROOF. The key computation cost is checking whether $R \cup \{u\} \cup \{v\}$ is a $s$-dclique (line 16 of Algorithm 2). As described above, the maintenance of $A$ takes linear time for each node $u \in C$. Thus, the total cost for maintaining $A$ is $O(|C|^2)$. With the help of the array $A$, computing $\overline{m}(v, R \cup \{u\})$ only needs constant time for each $v \in C$. Therefore, the total cost is also $O(|C|^2)$. Putting it all together, the total time complexity on each recursion node is $O(|C|^2) = O(\delta^4)$. □