# OpenReview forum: "Counting Cohesive Subgraphs with Hereditary Properties"
_ACM.org/TheWebConf/2025/Conference — WWW 2025 Poster_

### Official Review · Reviewer_Z5wp · 2024-11-23

**Novelty:** 6
**Technical Quality:** 6

**Review:**

The authors of this work essentially propose a generalization of the clique counting problem. They allow for a more general class of subgraphs that have the cohesive and hereditary properties, which as far as I can tell are "near-cliques".

The authors present algorithms for this problem and justify them with theoretical analyses. Then they conduct an extensive experimental evaluation against reasonable baselines and show the efficacy of their algorithms.

I see this as a well written paper that studies a nice problem. Clique counting has been studied extensively and here the authors study natural generalizations. The algorithms are novel enough to be interesting and the theoretical justifications are a plus. The experimental analysis is also convincing as far as I can see. Overall this is a good paper that I would recommend for acceptance.

**Questions:**

N/A

**Reviewer Confidence:**

3: The reviewer is confident but not certain that the evaluation is correct

**Scope:**

4: The work is relevant to the Web and to the track, and is of broad interest to the community

---

### Official Review · Reviewer_Y9Bb · 2024-12-02

**Novelty:** 6
**Technical Quality:** 5

**Review:**

This paper addresses the problem of counting general hereditary cohesive subgraphs (HCS), which extends the classic clique model by relaxing its restrictive conditions while preserving heredity and cohesiveness. The authors propose the HCSPivot framework, which efficiently counts HCS combinatorially without explicitly listing them. Two novel algorithms are introduced to count specific types of HCS, and extensive experiments demonstrate their high efficiency and effectiveness. I suggest to accept this paper. Here are some concrete comments:
1. The innovation in this paper provides certain inspiration in the direction of this field;
2.The research methods of the paper are clearly introduced, which can clearly prove the correctness of the ideas put forward.
3. The tables and figures in this article are clear and unmistakable.

**Questions:**

Can this method be extended to arbitrary values of q and s?

**Reviewer Confidence:**

3: The reviewer is confident but not certain that the evaluation is correct

**Scope:**

4: The work is relevant to the Web and to the track, and is of broad interest to the community

---

### Official Review · Reviewer_K7x4 · 2024-12-02

**Novelty:** 5
**Technical Quality:** 4

**Review:**

This manuscript addresses the challenge of counting Hereditary Cohesive Subgraphs (HCS) and introduces a pivot-based framework called HCSPivot. The framework integrates a range of innovative pruning techniques and algorithmic designs to tackle various HCS counting problems, such as s-defective cliques and s-plexes. Notably, this is the first method specifically designed for HCS counting. Experiments conducted on eight real-world network datasets demonstrate the approach’s effectiveness.

Cons:
1) The discussion in Section 3 is somewhat brief and does not provide a deep analysis of key design choices, such as the candidate set partitioning strategy and the pruning techniques.
2) Although the authors present quantitative comparisons of runtime and pruning rates, the paper lacks a thorough discussion on the robustness and scalability of the proposed method.
3) The algorithm’s complexity might make it challenging for readers unfamiliar with counting problems. Including a toy example in Section 3 could make the framework more accessible and easier to understand.
4) The experiments compare the proposed method only with traditional and enhanced approaches developed by the authors, omitting mainstream algorithms that could provide a broader benchmarking perspective.

**Questions:**

1) In Section 4.3, the authors introduce two targeted pruning methods. Could you provide corresponding pseudocode or illustrative diagrams to enhance clarity? Additionally, since pruning introduces extra computational overhead, on what types of networks (dense or sparse) does it prove more effective?
2) The comparison experiments use only baselines proposed within this paper. Could you include comparisons with more recent enumeration algorithms or heuristic methods? Alternatively, how does the proposed method perform compared to state-of-the-art clique counting approaches?
3) Table 2 reports only the runtime. Could you include statistical results showing the counts obtained for different methods under various q values? Are the counting results consistent across the methods?
4) In Section 5.3, the clustering experiment is limited to the email-Eu-core network. Does the proposed HCS-based solution remain effective on denser and larger-scale networks? Additionally, how does it compare to other clustering algorithms in terms of runtime performance?

**Reviewer Confidence:**

3: The reviewer is confident but not certain that the evaluation is correct

**Scope:**

3: The work is somewhat relevant to the Web and to the track, and is of narrow interest to a sub-community

---

### Official Review · Reviewer_sT4Q · 2024-12-03

**Novelty:** 2
**Technical Quality:** 2

**Review:**

The paper proposed a novel pivot-based framework called HCSPivot to count hereditary cohesive subgraphs (HCS) in graphs, which addressed the limitations of the classic clique model. This framework can simultaneously count HCS of any size and HCS associated for each node or edge. Based on HCSPivot, the paper introduces two novel algorithms with pruning techniques to efficiently count two specific HCS types including s-defective cliques and s-plexes. Extensive experiments demonstrate the high efficiency and effectiveness of the proposed solutions.

**Questions:**

Strong Points:

S1. HCSPivot is the first framework designed to count hereditary cohesive subgraphs (HCS), motivated by limitations of the classic clique model in real-world applications. This framework has two practical applications in graph analysis, including motif-based graph clustering and network comparison.

S2. Clear review of current literature is provided.

S3. Experimental evaluations on multiple real-world datasets are provided.

Weak Points:

W1. The paper lacks clear explanations of the reasons for choosing current technical (D1).

W2. Limited technical innovation (D2).

W3. Lack of clear and concise examples (D3).

W3. Limited baselines in experiment setting (D4).

W4. The experimental section has deficiencies (D5, D6).

W5. Unclear Grammar and expression (D7, D8).

Detailed Evaluation:

D1. In Introduction part, the paper mentions various relaxed clique models, such as s-defective clique, s-plex, s-clique, γ-quasi-clique, k-core, k-truss, and k-edge connected subgraph, but it does not clarify why the study focuses solely on those with both hereditary and cohesive properties are considered.

D2. The proposed framework employs simple strategies such as pivot-based approach and pruning techniques.

D3. The paper discusses potential applications of the HCSPivot framework but it lacks clear, concise examples demonstrating these applications in practical scenarios.

D4. The current evaluation focuses primarily on comparing HCSPivot with Dlist and Plist, which are internal baselines or simpler listing-based methods. This limited scope may not fully showcase relative advantages in diverse or more challenging scenarios.

D5. In experiment result shows that the performance varies significantly with different parameter settings, particularly s-values in s-defective cliques or s-plexes, but does not provide a solution to select a proper value. This might raise concerns about the general applicability and robustness of the proposed HCSPivot framework.

D6. In experiment part, the paper does not provide an analysis of how different s-values impact memory usage in the HCSPivot framework. The paper does not mention whether there are additional optimization opportunities for improvement based on listing strategy.

D7. In part 2, the sentence “The neighbors of each node … is…” has a grammar error. It should be “are”.

D8. In part 3.2, the sentence “By hereditaries, this sub-part is also an HCS.” has a grammar error.

**Reviewer Confidence:**

3: The reviewer is confident but not certain that the evaluation is correct

**Scope:**

3: The work is somewhat relevant to the Web and to the track, and is of narrow interest to a sub-community